# Visible Light Reductive Photocatalysis of Azo-Dyes with n–n Junctions Based on Chemically Deposited CdS

**DOI:** 10.3390/molecules27092924

**Published:** 2022-05-04

**Authors:** Michele Mazzanti, Martina Milani, Vito Cristino, Rita Boaretto, Alessandra Molinari, Stefano Caramori

**Affiliations:** Dipartimento di Scienze Chimiche, Farmaceutiche ed Agrarie, Università di Ferrara, Via Luigi Borsari 46, 44121 Ferrara, Italy; michele.mazzanti@unife.it (M.M.); martina.milani@unife.it (M.M.); vito.cristino@unife.it (V.C.); rita.boaretto@unife.it (R.B.)

**Keywords:** TiO_2_, CdS, n–n junctions, photocatalysis, photoreduction, visible light, azo-dye

## Abstract

New composite photocatalysts have been obtained by chemical bath deposition of CdS on top of either nanostructured crystalline ZrO_2_ or TiO_2_ films previously deposited on conductive glass FTO. Their morphological, photoelectrochemical and photochemical properties have been investigated and compared. Time resolved spectroscopic, techniques show that in FTO/TiO_2_/CdS films the radiative recombination of charges, separated by visible illumination of CdS, is faster than in FTO/ZrO_2_/CdS, evidencing that carrier dynamics in the two systems is different. Photoelectrochemical investigation evidence a suppression of electron collection in ZrO_2_/CdS network, whereas electron injection from CdS to TiO_2_ is very efficient since trap states of TiO_2_ act as a reservoir for long lived electrons storage. This ability of FTO/TiO_2_/CdS films is used in the reductive cleavage of N=N bonds of some azo-dyes by visible light irradiation, with formation and accumulation of reduced aminic intermediates, identified by ESI-MS analysis. Needed protons are provided by sodium formate, a good hole scavenger that leaves no residue upon oxidation. FTO/TiO_2_/CdS has an approximately 100 meV driving force larger than FTO/ZrO_2_/CdS under illumination for azo-dye reduction and it is always about 10% more active than the seconds. The films showed very high stability and recyclability, ease of handling and recovering.

## 1. Introduction

Textile and other industrial dyes are one of the largest classes of organic compounds to represent a growing environmental issue due to the fact of their toxicity. In addition, their chemical structures make them stable and difficult to degrade. Azo-dyes constitute the large majority (50/70%) of this class of pollutants and they are the main type of industrial dyes current available in worldwide markets. Many azo-dyes display carcinogenic and mutagenic effects, with harmful impacts both on environmental and human health [1,2,3]. For the removal of these pollutants from wastewaters various methods have been proposed: biodegradation, coagulation, membrane processes, adsorption, and advanced oxidation processes (AOPs) [4,5,6,7,8,9]. Among the ones proposed, semiconductor photocatalysis has been widely applied. The literature reports about the ability of several wide bandgap metal oxides, such as titanium dioxide (TiO_2_) [10,11,12,13,14] or zinc dioxide (ZnO) [15], to induce oxidative degradation of organic pollutants by light irradiation. It is generally accepted that the photocatalytic degradation process is promoted by light irradiation that generates electron/hole pairs in the semiconductor. Their reaction with O_2_ and hydroxyl surface groups leads to the formation of reactive oxygen species (ROS), such as ^•^OH radicals and ^•^O_2_^−^, that can initiate the degradation of the pollutant. Even though photocatalysts are effective in achieving dye degradation, the mineralization of its oxidation-intermediates is slow and often incomplete even after long irradiation time [14].

Photocatalysis with semiconductors can make it possible to degrade a pollutant through reduction processes involving electrons in the conduction band. Concerning azo-dyes, some of us recently reported that, upon UV illumination, TiO_2_ was able to catalyze the reductive cleavage of N=N bond in the presence of sodium formate as holes scavenger [16]. However, one of the main drawbacks of TiO_2_ is its inefficient exploitation of visible light. Strategies to improve photocatalytic performances and to shift light absorption to the visible region are one of the main purposes of the most recent literature [17,18,19]. In particular, the intimate combination of at least two materials results in the formation of heterojunctions. An important consequence is that recombination of separated charges is slowed down if they are allowed to move on two different materials. Spatial separation inhibits recombination and lengthens lifetimes, thereby promoting redox processes. Therefore, in this paper we focus on new composite photocatalysts where TiO_2_ is coupled with cadmium sulfide (CdS), a visible light active photocatalyst with a bandgap of 2.4 eV which enables to harvest green and blue visible light. CdS itself has remarkable carrier transportation capability, which makes photoproduced electrons and holes movable in a convenient and efficient way [17]. We deeply investigate the fabrication of an appropriate heterojunction between CdS and TiO_2_ showing that the lifetimes of the separated electron-hole pairs are longer and that this is a convenient approach to exploit the enhanced charge separation and achieve higher photocatalytic performance. In addition, since energetics of the bands of TiO_2_ and CdS is suitable for electron injection from photoexcited CdS to dark TiO_2_ [20] and, in principle, TiO_2_/CdS composite system should be able to perform the photocatalytic transformations previously explored for UV/TiO_2_ but using visible light and CdS as the unique photoactive material. For this, the reductive cleavage of N=N bond in three representative azo-dyes in the presence of formate (already studied with UV/TiO_2_) will still be the process of interest.

The composite photocatalyst will be deposited in the form of nanocrystalline thin film on FTO (Fluorine Tin Oxide) (FTO/TiO_2_/CdS). Its photoelectrochemical and photo-chemical properties will be deeply investigated and the formation and the effect of the heterojunction between the two oxides will be evaluated. A comparison with FTO/ZrO_2_/CdS films, where zirconia is used in place of titania will be carried out. In this last system, in contrast with FTO/TiO_2_/CdS, the electron transfer from illuminated CdS to ZrO_2_ is energetically precluded, due to the high bandgap of zirconia.

The use of thin films deposited on FTO circumvents all the difficulties related to the use of slurries. Moreover, the photostability of CdS can be significantly improved, opening to the use of this colored material as a photocatalyst. In addition, immobilized films are easy to handle, recover and recycle.

## 2. Materials and Methods

### 2.1. Materials

HCOONa (Sigma Aldrich, St. Louis, MO, USA, 99.5%), ethanol (Fluka, St. Louis, MO, USA, >99.8%, UV grade), CdCl_2_ (Alfa-Aesar, Karlstuhe, Germany, 99%), KOH (Lancaster, Morecambe, England, 85–90%), NH_4_NO_3_ (Riedel-de Haën, Hannover, Germany, 99%), CS(NH_2_)_2_ (Alfa-Aesar, Haverhill, US, 99%), LiClO_4_ (Acros Organics, Jeel, Belgium, >99%), TiO_2_ colloidal paste (Dyesol 18NRT, Queenbeyan, Australia), zirconium(IV) *n*-propoxide (70% *w/w n*-propanol, Alfa Aesar, Karisruhe, Germany), polyethyleneglycol, bisphenol A, epichloridrin copolymer (Sigma, Carbowax), and ZrO_2_ were prepared according to procedures in the literature [21]. Methyl orange (MO, Carlo Erba Reagents, Milan, Italy >99.98%) and acid orange 7 (AO7, VWR, Milan, Italy, ≥97%) were purchased and used without further purification. Their structures are reported in Figure 1.

### 2.2. Synthesis of Ethyl Diazosalycilate (EDS)

EDS was synthesized following the procedure described below: sulfanilic acid (15.5 mmol, Sigma Aldrich, St. Louis, MO, USA) and Na_2_CO_3_ (6.25 mmol, Carlo Erba Reagents, Milan, Italy) were dissolved in water (25 mL). The solution was cooled at 15 °C and NaNO_2_ (14.7 mmol, Carlo Erba Reagents, Milan, Italy) was added. The as obtained solution was slowly added to a frozen one of commercial HCl (37% *w/w*, Sigma Aldrich, St. Louis, MO, USA). The precipitation of diazobenzenesulfonate is observed. The copulation reaction with salicylic acid was carried out by mixing the above acid solution with a basic (NaOH 10%) aqueous solution containing salicylate (12.5 mmol, Sigma Aldrich, St. Louis, MO, USA). As a result of the addition, the dye immediately formed as a solid product, which re-dissolves in the reaction environment upon prolonged stirring and heating. After complete dissolution of the dye, NaCl (85 mmol, Sigma Aldrich, St. Louis, MO, USA) was added causing the precipitation of (3-(carboxy)-4-(hydroxyphenyl)diazenyl)benzenesulfonate as a sodium salt. The precipitated orange-brown dye was recovered from the solution by filtration and subsequently washed with a saturated NaCl. The esterification reaction was carried out by reacting the crude dye (0.62 mmol) with ethanol (10 mL) in the presence of H_2_SO_4_ (0.1 mL 95–97%, Merck, Darmstadt, Germany) at 70 °C for 19 h. Ethyl diazosalicylate (EDS) was identified by ^1^H-NMR. Observed signals refers to a mixture of sin and anti diastereoisomers (8.01 ppm (1H, d, *J* = 2.5 Hz) anti; 7.88 ppm (1H, d, *J* = 2.5 Hz) sin; 7.80 ppm (4H, m) sin; 7.71 ppm (1H, d-d, *J* = 8.5; 2.5 Hz) anti; 7.62 ppm (1H, d-d, *J* = 8.5; 2.5 Hz) sin; 7.58 ppm (2H, d, *J* = 8.0 Hz) anti; 7.52 ppm (2H, d, *J* = 8.0 Hz) anti; 6.82 ppm (1H, d, *J* = 8.0 Hz) anti; 6.78 ppm (1H, d, *J* = 8.5 Hz) sin). Its structure is reported in Figure 1.

### 2.3. Preparation of FTO/TiO_2_ and FTO/ZrO_2_ Thin Films

Porous TiO_2_ films were obtained by blade casting a commercial terpineol based paste (Dyesol 18NRT, Queenbeyan, Australia) on well cleaned FTO, according to previously published directions [16]. ZrO_2_ was similarly obtained by spreading the ZrO_2_ paste on FTO, followed by drying at room temperature and sintering at 500 °C in air.

### 2.4. Preparation of FTO/ZrO_2_/CdS and FTO/TiO_2_/CdS

Modification of either ZrO_2_ or TiO_2_ thin films was obtained by chemical bath deposition (CBD) by slow hydrolysis of thiourea in the presence of Cd^2+^: the aqueous chemical bath was composed of 80 mL of CdCl_2_ (0.02 M), 200 mL of KOH (0.5 M), 80 mL of NH_4_NO_3_ (1.5 M) and 80 mL of CS(NH_2_)_2_ (0.2 M) giving 440 mL as total volume, with pH = 11. The reaction occurred under heating with a hot-plate at 80 ± 1 °C for 30 min and the thin films were kept immersed vertically in the chemical bath. The FTO contacts were protected with Kapton tape.

### 2.5. Structural Characterization of the Semiconductor Films

The films thicknesses were measured using an Alpha step D-500 Profilometer (KLA instruments, Milipitas, CA, USA). Data were obtained in step-up/down mode with a scan length of 3.5 mm at a speed of 0.07 mm/s and a stylus force of 5.0 mg.

Atomic force microscopy (AFM) images were collected using a Digital Instruments Nanoscope III scanning probe microscope (Veeco-Digital Instruments, Plainview, NY, USA). The instrument was equipped with a silicon tip (RTESP-300 Bruker, Billerica, MA, USA) and operated in tapping mode. Surface topographical analysis of AFM images was carried out with a NanoScope Analysis 1.5.

Scanning electron microscopy (SEM) of the films was obtained with a Zeiss EVO 40 scanning electron microscope apparatus (Zeiss, Jena, Germany).

X-ray diffraction (XRD) measurements were performed using a BRUKER D8 Advance X-ray diffractometer (Bruker, Billerica, MA, USA) equipped with a Sol-X detector, working at 40 kV and 40 mA. The X-ray diffraction patterns were collected in a step-scanning mode with steps of Δ2θ = 0.02° and a counting time of 10 s/step using Cu Kα1 radiation (λ = 1.54056 Å) in the 2θ range of 3–80° using an incident grazing angle set-up.

### 2.6. Steady State Optical Absorption and Emission

Absorption Spectra of the semiconductor films were collected under diffuse reflectance (R%) mode with a JASCO V 570 spectrophotometer (JASCO, Tokio, Japan) equipped with an integrating sphere. Tauc plots were obtained according to (F(R) × hν)^α^ = A × (hν − E_g_) where α = 1/2 for a direct band gap, (F(R) = (1 − R)^2^/2R) is Kubelka-Munk function, A is a proportionality coefficient and E_g_ is the semiconductor band gap. Emission spectra (λ_exc_ = 450 nm) of the thin films at room temperature were obtained with an Edinburgh Instruments FLS 920 spectrofluorometer (Edinburgh Instrument Ltd., Livingston, UK) using a dedicated film holder. Both emission and excitation slits were set at 8.0 nm during these measurements. Spectra were corrected for the lamp and photomultiplier response and averaged over 10 subsequent scans with a 1 nm step.

### 2.7. Electrochemistry and Photoelectrochemistry

Electrochemical and photoelectrochemical measurements were carried out with a Metrohm Autolab PGSTAT 302/N electrochemical workstation (Methrom Autolab, Utrecht, The Netherlands). The redox properties of MO, AO7 and EDS dyes were investigated by cyclic voltammetry in a three electrode GC (glassy carbon)/Pt/SCE (Saturated Calomel Electrode) cell in aqueous 0.1 M LiClO_4_ at 50 mV/s. 

Photoelectrochemical experiments were collected under solar simulated illumination with an ABET Sun Simulator (ABET Technologies, Milford, CT, USA) equipped with AM 1.5 filter. The incident irradiance was set to 0.1 W/cm^2^ with a Newport Power Meter model 1918-c. Pulsed illumination experiments were obtained with a Thorlabs electronic shutter. All photoelectrochemical experiments were carried out in a three electrode cell by using aqueous 0.1 M HCOONa as electrolyte. A SCE was the reference electrode, a Pt wire was used as a counter electrode, either FTO/TiO_2_/CdS or FTOZrO_2_/CdS films were used as the working electrodes. J/V (Current Density vs. Voltage) curves of the photoelectrodes were recorded by cyclically scanning the voltage between 1 and −1 V vs. the SCE at a scan rate of 20 mV s^−1^ both under AM 1.5 G and under dark conditions. Open circuit chronopotentiometry was performed as follows: initially the photoanode was positively polarized in the dark at 0.5 V vs. SCE for 100 s and then allowed to reach a stable potential in the dark. Usually, after 80s the dark potential reaches a steady value and the substrate is irradiated with AM1.5 light, causing generation of electrons and holes, which may recombine or undergo separation and storage within the semiconductor. Illumination of the photoelectrode is maintained until a stable value of the photopotential is obtained. After this was achieved, restoration of the dark conditions causes a decay of the photovoltage owing to recombination, allowing to extract the electron lifetimes as a function of the Fermi energy of the photoelectrode.

### 2.8. Photoaction Spectra

Photoaction spectra (IPCE = Incident Photon to electron Conversion Efficiency) were collected under mono-chromatic illumination generated by the coupling of a Luxtel 175W Xe lamp to an applied photophysics monochromator (Luxtel, Danvers, MA, USA), using a spectral bandwidth of 10 nm and a constant potential of 0 V vs. SCE using the same three electrode and potentiostat described in Section 2.6. Incident irradiance was measured with a calibrated silicon photodiode while the short circuit photocurrents were measured with an Agilent 34401 A multimeter.

### 2.9. Single Photon Counting

Emission lifetimes of the CdS containing photoanodes immersed in aqueous electrolyte (1 M HCOONa) were acquired with a Picoquant Picoharp 300 time correlated single photon counting at a 4 ps resolution by using a 480 nm pulsed LED source. Levenberg-Marquardt fitting/deconvolution of the decay histogram was accomplished with a tri-exponential function by the dedicated Fluofit program. In general, fits satisfied the statistical acceptability criteria, with χ2 ≈1 and residuals R(i) = W(i)(Decay(i) − Fit(i)) < 4 standard deviations fluctuating around 0 within all the fitting intervals. In the R(i) formula W(i) = 1/(decay(i))1/2 defines the intensity weight in a given channel (i) according to the Poisson distribution while Decay(i) and Fit(i) are the experimentally measured and calculated decay values respectively.

### 2.10. Open Circuit Photocatalysis

Prolonged irradiation of the semiconductor thin films supported on FTO was carried out with an Oriel Xe/HgXe lamp. A glass cut-off filter was used (λ > 420 nm).

Typically, a FTO/TiO_2_/CdS (or FTO/ZrO_2_/CdS) sheet was placed inside a Pyrex cell in front of the optical face and immersed in an aqueous solution (3 mL) containing HCOONa (1 M) electrolyte and the dye of interest (MO, AO7, or EDS, C_0_ = 10 ppm). The Pyrex cell was closed and degassed by N_2_ bubbling for 30 min, then the semiconductor film was front (through CdS) illuminated. The dye degradation kinetics were monitored by UV-visible spectrophotometry in the 200–600 nm interval by using a Cary 300 UV-vis double beam spectrophotometer (Agilent Technologies, Santa Clara, CA, USA) from which the dye concentration was obtained according to the Lambert-Beer law. Dye decay curves were plotted as C/C_0_ (C_0_ = initial dye concentration) vs. the irradiation time.

Catalyst stability was evaluated by employing the FTO/TiO_2_/CdS and FTO/ZrO_2_/CdS sheets in several irradiation cycles of 60 min duration, as described above. Between two subsequent cycles, the electrodes were rinsed with deionized water and dried in air at room temperature. During these experiments, the irradiated solutions were periodically subjected to Atomic Absorption Spectroscopy (AAS) in order to evaluate Cd^2+^ release from the sheet.

### 2.11. Atomic Absorption Spectroscopy (AAS)

AAS was carried out using an atomic absorption spectrophotometer with an electrothermal atomizer (transverse heating graphite tube), Analyst 800 model by Perkin Elmer (Waltham, MA, USA) equipped with an AS91 model autosampler. The calibration curve was obtained from seven standard cadmium solutions, prepared by dilution of a certified cadmium standard of 1000 µg/L; the analyzed samples were adequately diluted to enter the linearity of the curve.

### 2.12. ESI–MS

Mass spectra of the dye solution subjected to photocatalysis were recorded using an LCQ Duo (ThermoQuest, San Jose, CA, USA), equipped with an electrospray ionization source (ESI), monitoring the precursor-to-product ion transitions of *m/z* 100 to 400 in the negative ionization mode. In order to accomplish ESI-MS analysis without interference of ionic species, photocatalysis was carried out with ethanol (10% *v/v*) instead of HCOONa as a nonionic hole scavenger.

## 3. Results and Discussion

### 3.1. FTO/TiO_2_/CdS and FTO/ZrO_2_/CdS Thin Films: Structural and Optical Properties

Chemical bath deposition (CBD) of CdS on top of either nanostructured ZrO_2_ or TiO_2_ films (MO_2_ films, where M is either Ti(IV) or Zr(IV)) affords a visually homogeneous orange film (Inset in Figure 1a,b). The overall thickness of the ZrO_2_/CdS and TiO_2_/CdS layers was of the order of 4 and 8 µm respectively (Appendix A). No significant increase in the film thickness was observed prior and after deposition of CdS, meaning that we form an interpenetrated junction where the electrolyte, during CBD depositions, percolates within the TiO_2_ or ZrO_2_ mesopores and is conformably deposited on the nanoparticle surface. The SEM imaging reveals a homogeneous network of nanoparticles (Figure 1a,b), which can be better resolved with AFM (Appendix A), where spheroidal particles of the approximate size of 25 nm are observed on ZrO_2_, while large structures having with a ca. 50 nm diameter are seen on TiO_2_. Thus, after CdS deposition the junctions maintain their mesoporous nature, with pores and voids allowing electrolyte permeation.

XRD of a pure CdS sample (Appendix A) deposited on FTO by CBD does not reveal any diffraction pattern originating from CdS, but only the obvious presence of crystalline FTO giving rise to intense and sharp peaks according to the well crystallized cassiterite structure. XRD of the n–n junctions supported on FTO display a complicated pattern due to several overlapping contributions from FTO and from the nanoparticulate crystalline phases of TiO_2_ (anatase [101] at 2θ = 26°) and ZrO_2_ ([101], at 2θ = 31° and [201] 2θ = 51°) particularly within the 2θ interval between 25 and 30° (Appendix A). We can however identify clear diffraction peaks from CdS at 2θ = 44° [220] and 2θ = 52° [311], in a region relatively free from interfering patterns originated by the other compounds [22,23]. This indicates that the CBD procedure produces a completely amorphous CdS layer on bare FTO, but the presence of the underlying crystalline structure of ZrO_2_ and TiO_2_ induces the formation of crystalline CdS domains. The broad diffraction peaks of CdS are however indicative of a small size of the coherent diffraction domains and of a substantial disorder of the CdS semiconductor film grown by CBD on top of both TiO_2_ and ZrO_2._ The absorption spectra of the CdS modified MO_2_ films from diffuse reflectance data in Kubelka Munk (KM) units (Figure 2) are consistent with the presence of CdS displaying a direct band gap of 2.15 ± 0.3 eV, extracted from Tauc Plots (Appendix A). From the raw reflectance data, we observe that the light-harvesting efficiency between 400 and 500 nm, where CdS manifests its absorption peak, is ca. 90%.

CdS modified films were found emissive at room temperature upon 450 nm excitation. Compared to a pure CdS film deposited on bare FTO (FTO/CdS), both the FTO/MO_2_/CdS films display similarly broadened emission spectra, with a red shift in the emission maximum of approximately 20 nm (Appendix A). The broadened and red shifted response might be the result of increased heterogeneity in the nanoparticulate films and of some stabilizing interaction between CdS and the mesoporous oxide, which lower the exciton energy. Time resolved analysis of the fluorescence revealed a multiexponential decay, consistent with a distribution of sites on the MO_2_/CdS from which the radiative recombination occurs with different rate constants. Typically, the emission decays (Appendix A) were satisfactorily fitted with a triexponential function, where the largely dominant (≈98%) component is below the instrumental resolution of our apparatus (<300 ps), while smaller amplitudes (≈1–2%) were much longer lived and decayed in the nanosecond time scale. Interestingly, the ns radiative recombination was faster on FTO/ZrO_2_/CdS where amplitudes of 0.48 and 0.31% decayed with a lifetime of 4.01 and 1.44 ns (integrated intensities 15.94 and 3.71, respectively, Appendix A) compared to FTO/TiO_2_/CdS, where amplitudes of 1.04% and 0.11% exhibited a lifetime of 2.52 and 12.9 ns (integrated intensities of 16.5% and 9.15%, respectively, Appendix A). This indicates that already on such short time scales, the radiative recombination of the carriers is slowed down on TiO_2_, probably due to the injection of electrons from the CdS phase to anatase and subsequent recombination occurring at the interface between these two materials (Figure 2).

We note that the observation of the kinetics on the ns time scales does not allow to draw sure conclusions about the comparative photocatalytic activity of the FTO/MO_2_/CdS films under investigation, given that usually multi-electron heterogeneous processes relevant to photocatalysis occur on much longer time scales (milliseconds/seconds) [24] and often involve trapped long lived carriers that do not recombine radiatively. Nevertheless, it provides a first indication of different carrier dynamics in the two systems. Photoelectrochemical experiments will be used to gain further indication of a longer lived charge separation in FTO/TiO_2_/CdS.

### 3.2. FTO/TiO_2_/CdS and FTO/ZrO_2_/CdS Thin Films: Photoelectrochemical Properties

Photoelectrochemical investigations provide a comprehensive picture of the charge separation and recombination dynamics of FTO/MO_2_/CdS films in the same electrolyte and similar illumination conditions to those used for photocatalytic experiments. In particular, the photocurrent density and photovoltage decay are related to the efficiency of charge separation, which ultimately affects the photocatalytic performance towards azo-dyes reduction. The photocurrent density/voltage curves of the FTO/MO_2_/CdS in 1 M HCOONa are reported in Figure 3 along with the photoaction spectra (IPCE vs. λ) recorded under 0 V vs. SCE bias.

The J/V curves are consistent with a n-type junction where TiO_2_ offers an unimpeded electron collection to the electrons generated upon white light illumination of CdS/TiO_2_ junction. A photocurrent plateau, extending from +1 and −0.25 V vs. SCE is observed, with a photovoltage close to −1 V vs. SCE. A similar figure, both in terms of photocurrent and photovoltage, is observed for FTO/CdS. A similar photovoltage is observed in the case of FTO/ZrO_2_/CdS, which, however, delivers a much lower photocurrent (ca. 0.1 mA/cm^2^). This agrees with the prohibitively high conduction band edge of ZrO_2_ which precludes electron injection from CdS. Here ZrO_2_ behaves as an inert insulating substrate, which only offers support for nucleation of CdS during the CBD growth. Collection of charge is thus possible only through the random direct contact between CdS and FTO, hence the electron collection through the ZrO_2_/CdS network is considerably suppressed. Consistent with this interpretation, we observe maximum photon to electron quantum yields (measured at the photocurrent plateau value at 0 V vs. SCE) of 50–55% for FTO/TiO_2_/CdS > FTO/CdS (40–45%) >> FTO/ZrO_2_/CdS (2–2.5%, Figure 3c inset). The significant quantum yield detected with FTO/TiO_2_/CdS also confirms that HCOO^−^ behaves as an efficient hole scavenger, leading to a >60% conversion of absorbed photons into electrons flowing through the electrochemical cell. Dark cyclic voltammetry (−1 V/1 V vs. SCE) of the FTO/TiO_2_/CdS (Appendix A) reveals two subsequent quasi-reversible waves, a first less intense process centered at approximately −0.6 V vs. SCE, assigned to filling of trap states of TiO_2_, followed, at stronger negative polarization, by a second more intense process, assigned to reduction of a higher density of electron acceptor states, probably conduction band states. By contrast CVs of FTO/CdS and FTO/ZrO_2_/CdS (Appendix A respectively) exhibit similar features assigned to the irreversible reduction of CdS giving rise to a continuous discharge with a shoulder around −0.7 V vs. SCE. This means that a driving force of at least ≈ 100 meV exists for the transfer of photoexcited electrons of CdS to TiO_2_ and that trap states of TiO_2_ can act as a reservoir for long lived charge storage. Photovoltage decay experiments allow to observe the recombination dynamics of long lived charge carriers which are relevant to photocatalysis. First, we observe that under illumination in 1 M sodium formate, the photovoltage of FTO/TiO_2_/CdS is ca. 100 mV more negative (−0.9 V vs. SCE) than that of FTO/ZrO_2_/CdS (≈ −0.8 V vs. SCE) and that the photovoltage decay upon restoration of the dark conditions occurs with a slower kinetics in the former (Appendix A). Electron lifetime can be obtained from the open circuit photovoltage decay (V = (E*_F_ − E_SCE_)/(−e), where –e is the electronic charge, E*_F_ is the quasi Fermi energy of the photoanode and E_SCE_ is the reference potential) as a function of the voltage as [25]:(1)$$\tau =\frac{kT}{e}{\left(\frac{\partial V}{\partial t}\right)}^{-1}$$

The resulting plot is reported in Figure 4, from which we can observe that the electron lifetime in FTO/TiO_2_/CdS is almost two orders of magnitude longer than in FTO/ZrO_2_/CdS within the potential range −0.9/−0.6 V vs. SCE. This is a further confirmation of transfer and storage of electrons from CdS to the TiO_2_ phase. Based on this experimental piece of evidence we expect the FTO/TiO_2_/CdS films to be able to outperform FTO/ZrO_2_/CdS films in terms of long lived charge separation ability.

Reduction of two out of three of the azo-dyes (MO and AO7) under consideration occurs according to an irreversible process centered at ca. −0.8 V vs. SCE and is thus exoergonic by approximately 100 meV with respect to the quasi Fermi potential achieved by FTO/TiO_2_/CdS, whereas it is almost isoergonic with that of FTO/ZrO_2_/CdS. Reductive cleavage of EDS should be thermodynamically easier with both photocatalysts, according to its irreversible wave at −0.4 V vs. SCE (Appendix A). The differential capacitance of the semiconductor films *C* ($C=\frac{dQ}{dV}$, where *Q* is the charge and *V* the applied voltage) is obtained from the voltammetric curves of the junctions in 1 M sodium formate according to *C = i/v,* where *i* is the current and *v* is the scan rate, which is reported in Figure 5, vs. the applied potential (−E_F_/e), together with the relevant redox levels for the reductive chemistry under consideration. *C* is proportional to the electronic density of states *DOS* (variation of the number of electronic states *N* as a function of the Fermi energy) according to:(2)$$DOS=\frac{dN}{dE_{F}}=\frac{C}{-e^{2}}$$

By comparing C_CdS_ with C_TiO2/CdS_, we see that electron transfer from CdS to TiO_2_ will be efficient, in agreement with the photoelectrochemical evidence discussed before. This is due to the fact that electronic states of CdS and TiO_2_ have a large energy overlap. We also observe that the deposition of CdS on ZrO_2_ has a quite minor impact on the energy distribution of CdS states: C_CdS/ZrO2_ only tends to be slightly larger than that of pure CdS at lower potential (<−0.6 V vs. SCE). We note that we limited the potential scan to −1 V vs. SCE, due to the interception of electrolyte reduction occurring beyond that value. Nevertheless, such potential range embraces the maximum open circuit value recorded under our conditions. Electronic transfer from CdS to TiO_2_, resulting in the confinement of electrons and holes on different phases, retards recombination, as evidenced from photovoltage decay analysis and improves the electronic build up in the n–n junction. As a result, the quasi-Fermi potential of FTO/TiO_2_/CdS under illumination increases to ca. −0.9 V vs. SCE. At such equilibrium potential, the overlap between the TiO_2_ and CdS states is very large, since the TiO_2_ capacitance increases very rapidly above −0.6 V vs. SCE, and we expect that electrons will be able to cross back (from CdS to TiO_2_) and forth (from TiO_2_ to CdS) the boundaries between CdS and TiO_2_ with a negligible activation energy. This will result in the population of both CdS and TiO_2_ states under dynamic equilibrium and in the possibility to drive simultaneously the azo-dye reduction both at the surface of CdS and of exposed TiO_2_ particles, making irrelevant the fact that in the FTO/TiO_2_/CdS architecture TiO_2_ is substantially covered by CdS, a fact that, at first glance, may preclude an efficient contact between the photocatalyst and the molecular electron acceptors dissolved in the electrolyte.

### 3.3. Open Circuit Photocatalysis

An FTO/TiO_2_/CdS slide, having a CdS coated area of 1 cm^2^, immersed in a deaerated aqueous solution containing the dye of interest (MO, AO7 or EDS, C_0_ = 10 ppm) and HCOONa (1 M), was irradiated from the CdS side (λ > 420 nm). This illumination geometry is referred to as front illumination mode. FTO/TiO_2_/CdS slide displays a nearly quantitative light absorption and from AFM measurements (see Section 3.1) we estimated that its surface area is much larger (by a factor of 10^4^) than that required to saturate the surface with dye molecule present in our photocatalytic cell.

During irradiation, a loss of the solution color that corresponds to a decrease of the absorption band of the azo-dye is observed. Figure 6 reports the C/C_0_ ratio of MO, AO7, or EDS as a function of irradiation time.

It is observed that after 180 min irradiation about 90% of the starting MO and AO7 has disappeared from the solution (full circles and triangles), while the decrease of C/C_0_ relative to EDS is slower and is around 70% despite the larger driving force that we foresee for such process. Control experiments (in Figure 6 are reported only those relative to MO) emphasize that this result is exclusively ascribable to the photocatalytic activity of CdS in the FTO/TiO_2_/CdS system. In fact, visible irradiation of the dye solution in the absence of any semiconductor (empty triangle in Figure 6) did not result in a dye concentration decrease. In addition, no decrease was observed when FTO/TiO_2_/CdS is placed in contact with the dye solution and kept in the dark (empty rhombus in Figure 6), also indicating that adsorption of MO on FTO/TiO_2_/CdS was negligible. Finally, irradiation (λ > 420 nm) of FTO/TiO_2_ system immersed in the MO solution did not cause any photosensitization effect on the dye, since TiO_2_ is not photochemically active above 420 nm (data not shown but superimposed with empty rhombus).

Considering that fading of the dye solution is only an indication of the interruption of conjugation and that monitoring the disappearance rate of the target dye is not the most appropriate way to establish the nature of the photocatalytic reaction, we recorded the negative ion mode ESI-MS spectrum of the irradiated solutions. Figure 7 reports the spectrum obtained in the case of AO7, while the analogous spectra for MO and EDS are reported in the Appendix A.

Moreover, the *m/z* peak at 327 related to the residual AO7, the dominant base peak was at *m/z* 172. The same result was obtained also with the other two investigated dyes (Appendix A). In agreement with our previous work [16], *m/z* 172 corresponds to the anionic form of sulfanilic acid, which is the common part in the structures of the studied dyes (Figure 1).

These results give evidence that the N=N bond undergoes a reductive cleavage (via protons and electrons addition). In fact, electron-hole pairs are generated on the surface of the composite material under visible light irradiation (Figure 3, Reaction (1)). Photogenerated holes effectively react with formate, which acts as a hole scavenger and as the proton source (Figure 3, Reaction (2)). The formation of CO_2_^−•^ radical has been recently evidenced by some of us [16]. At the same time, electrons promoted in the conduction band of CdS or those subsequently injected in that of TiO_2_ can be efficiently captured by dyes since we are operating in the absence of di-oxygen (Figure 3, Reaction (3)).

Compared to our previous work in which FTO/TiO_2_ plates were used [16], this result is a step forward in overall sustainability because we use visible light. We emphasize that the use of thin films on glass substrates circumvents all the difficulties related to the use of photocatalytic powder slurries; the films are easy to handle and to re-cover and no separation of the powder from the solution should be carried out at the end of irradiation. Usually, thin films also show excellent stability allowing very good recyclability [16].

Regarding the FTO/TiO_2_/CdS system studied here, we evaluated the photocatalytic performance in MO transformation in repeated cycles using the same plate. Figure 8 shows that there is no loss of photocatalytic activity after five repeated cycles, indicating a very good stability of the composite system.

Since it is known that the photostability of CdS is low, thus limiting its use as photo-catalyst, the result obtained with FTO/TiO_2_/CdS film is noteworthy and in line with the literature that reports that the photo-corrosion of CdS can be decreased when coupling with another semiconductor is carried out [26,27,28]. Atomic absorption spectroscopy measurements of irradiated samples report a release of 220 μg/L of Cd^2+^ ions into the solution after each cycle. In any case, we emphasize that this loss does not adversely affect the photocatalytic activity as shown by the results in Figure 8.

In addition, Figure 8 highlights that the presence of the underlying TiO_2_ layer in the FTO/TiO_2_/CdS films also improves the adhesion of CdS to FTO. In fact, we could not evaluate either the photocatalytic performance or the recyclability in the case of FTO/CdS films due to the very poor adhesion of CdS on FTO.

It has been highlighted in Section 3.1 and Section 3.2 that TiO_2_ plays an important role in the composite system by increasing the lifetime of the charge separated state and improving electron build up in the junction. From the photocatalytic point of view, this could result in high photocatalytic efficiency. This effect was confirmed by the comparative analysis of the photocatalytic performance of FTO/ZrO_2_/CdS films where zirconia is used as an electrochemically inert (with respect to electron transfer from photoexcited CdS) nanostructured support for CdS. Irradiation (λ > 420 nm) of a FTO/ZrO_2_/CdS slide, with a CdS coated area of 1 cm^2^, immersed in a deaerated aqueous solution containing the dye of interest (MO, AO7 or EDS, C_0_ = 10 ppm) and HCOONa (1 M), led qualitatively to the same process discussed above for FTO/TiO_2_/CdS. A decrease of C/C_0_ ratio of the dye as a function of irradiation time is observed (Appendix A), and ESI-MS spectra showed the same peak at *m/z* 172 related to sulfanilic acid formation (Appendix A). Figure 9 reports comparative azo-dye decay kinetics observed with the two FTO/MO_2_/CdS systems. It is observed that after 180 min of irradiation, C/C_0_ values are about 8–10% higher than those observed for FTO/TiO_2_/CdS. In addition, kinetic constants reported in Appendix A relative to the dye disappearance are always lower with FTO/ZrO_2_/CdS than for FTO/TiO_2_/CdS. These results are consistent with the fact that FTO/TiO_2_/CdS achieves, under illumination a ca. 100 meV larger driving force for azo-dye reduction with respect to FTO/ZrO_2_/CdS and that the partial coverage of TiO_2_ by CdS does not preclude the photocatalytic performance of the junction, once the stationary state under constant illumination is achieved.

Similarly to the FTO/TiO_2_/CdS system, the presence of ZrO_2_ underlayer has a positive effect on adhesion of CdS. This result affects the possibility to recycle the same plate several times: 60 min irradiation (λ > 420 nm) of the same FTO/ZrO_2_/CdS system immersed in an aqueous solution containing the dye (10 ppm) during five repeated experiments showed no decrease of the efficiency (data not shown). If consideration is made for the thicker titania film compared to ZrO_2_, the overall Cd^2+^ release appears similar to that observed with TiO_2_ (around 200 μg/L), confirming that the release is sufficiently low not to adversely affect the photocatalytic activity.

## 4. Conclusions

This study demonstrated that it is possible to obtain for CBD a nanocrystalline film of a TiO_2_/CdS composite material on FTO. For comparison, an analogous FTO/ZrO_2_/CdS system is studied in parallel. No significant increase in the film thickness was observed prior and after deposition of CdS, meaning that an interpenetrated junction is formed. The junction maintains the mesoporous structure after CdS deposition, so allowing permeation of CdS into pores and voids. From XRD we observe that CdS film grown on top of TiO_2_ is amorphous in general, with some crystalline domain.

On short time scale, the radiative recombination of the carriers is slowed down on TiO_2_, with respect to ZrO_2_ probably due to the injection of electrons from the CdS phase to anatase and subsequent recombination occurring at the interface between these two materials.

Photoelectrochemical investigations that provide a comprehensive picture of the charge separation and recombination dynamics of FTO/MO_2_/CdS films, demonstrates that a driving force of at least ≈100 meV exists for the transfer of photoexcited electrons of CdS to TiO_2_ and that trap states of TiO_2_ can act as a reservoir for long lived charge storage, that are those useful in photocatalysis.

This larger driving force confers a comparatively superior photocatalytic performance of FTO/TiO_2_/CdS with respect to FTO/ZrO_2_/CdS for the azo-dye reductive cleavage of N=N bond. In fact, nearly complete photocatalytic conversion of the studied azo-dyes under visible (λ > 420 nm) illumination is observed. Among the formed aminic products, sulfanilic acid is one of the reagents needed in the synthesis of fresh azo-dyes. This is a rare example of a photocatalytic reductive transformation that turns a waste into useful product by using visible light. The use of composite thin films, that circumvent all the problems related to the use of slurries, improves photostability of CdS leading to a recyclable material. The observed adhesion of CdS is good, with a low release of Cd^2+^ ions. Considering all the positive results obtained, future work will be focused on the minimization of the release, so opening the way to new truly heterogeneous photocatalytic systems working upon visible light illumination.

## Data Availability

The data used to support the findings of this study are included within the article.

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
