# Peer review of "Visible Light Reductive Photocatalysis of Azo-Dyes with n–n Junctions Based on Chemically Deposited CdS"

_molecules, 2022, doi:10.3390/molecules27092924_

Round 1

Reviewer 1 Report

The proposed manuscript deals with preparation of two thin films CdS/TiO2 and CdS/ZrO2 deposited on FTO glass and comparison of their photophysical/photochemical properties using a set of (photo)electrochemical methods and a photodegradation of model compounds (three azo dyes). Although the materials are well known and were intensively studied in the past, the authors provide precise and valuable measurements of fundamental photophysical/electrochemical properties and connect them to photocatalytic measurements on model compounds. The interpretations are based on precise experimental results and the whole manuscript is clear and concise. In my opinion the results are worth to publish, but several following comments should be resolved. 

1) The authors write: "during CBD depositions percolates within the TiO2 or ZrO2 mesopores.." There is no measurements of the porosity of the thin films. How the authors know that they are mesoporous?

2) Also, the particle size of the films is different (AFM). And there is no measurements of the specific surface area of the films. As the surface area is probably different, it can affect the photocatalytic activity of the films. Please comment.

3) The authors write: "The overall thickness of the ZrO2/CdS and TiO2/CdS layers was of the order of 4 and 8 μm respectively." The thickness of the one film is twice higher than the latter. Can this affect the photocatalytic activity or the adsorption of the azo dye?

4) FTO/ZrO2/CdS delivers a much lower photocurrent and other analyses also suggest its significantly worse photochemical properties. But its photocatalytic performance is only 8-10% lower compared to FTO/TiO2/CdS, which is for me quite surprising that there is so small difference. Can the authors provide some explanation for a relatively good performance of the FTO/ZrO2/CdS system?

5) The authors write: "..since visible irradiation of the dye solution in the presence of the FTO/TiO2 system, or in the absence of any semiconductor did not result in a dye concentration decrease." So there is no photosensitization effect of the azo dye adsorbed on FTO/TiO2? Can the authors comment?

6) The authors provide the evidence that Cd2+ is released to the solution. From the technical and environmental point of view using CdS with highly toxic cadmium is not realistic in the application such as water treatment. In this term the authors should focus in their future work on some other material, which is more environmentally friendly. I find all the performed measurements very valuable and would be nice to see them on some new photocatalytic materials that have higher potential for environmental applications.

Reviewer 2 Report

Caramori and coauthors reported an interesting work CdS/TiO2 for photoreduction of azo-Dyes. The work is well-performed and well-written; however, the following concerns must be addressed so that I can reconsider it for publication.

  1. Figure 1 has low quality and looks like raw data. I cannot see the scale bar clearly. The figure 1b is also not clear, seems like the focus is not adjusted well. SEM image should be redo and the figure must be remade.

2.What are those white balls on Figure 1a?

  1. Could you give the TEM images of your materials? TEM is much more straightforward than SEM and AFM.
  2. The authors used film as photocatalysts. What are the advantages of film form? I feel like the powder or solution based photocatalysts is more promising compared to the film, as they are easier to process and operate, and have more reactive sites. In contrast, for the film, the reaction only happened at the solid-liquid interface. The authors should add some related discussion in their paper.
  3. Following the question 4, how to get the catalyst loading of the reaction if the photocatalyst is film.
  4. The authors should also discuss the advantages of the heterostructures and cite some recent work using heterojunction photocatalysis ( ACS Materials Lett. 2022, 4, 3, 464–471; Chemical science 2021, 12 (44), 14815-14825). The advantages such as enhanced charge separation, suppressed charge recombination and others should be discussed in introduction.
  5. XRD, for “CdS at 2θ = 44° [220] and 2θ = 52° [311]”, any references support? Or PDF card of XRD?
  6. The inserted figures in Figure 2 have low quality and hard to read. They must be revised.
  7. Figure S4, does the y axis really called “a.u.” What is the TIO2CdS? Such carless mistakes should not happen in a scientific paper. The authors must double-check the whole paper.
  8. Figure S4, I do not understand why a multiple emission are in the presence of the authors materials. What is the peak around 620 nm in TiO2-CdS and ZrO2-CdS?
  9. The authors gave the normalized PL in Figure S4. In principle, if the heterojunction formed as the authors proposed in Scheme 2, the emission quenching should be observed. Did authors observe that? If yes, please provide the data. If no, why?
  10. How did the authors get the energy levels of CdS, TiO2 and ZrO2 should be presented.
  11. Can authors draw the dye degradation mechanism in SI? This will help the readers understand what is the final products and reaction process.
  12. Why did authors add HCOONa?
  13. Some control experiments such as FTO/CdS for photoreduction are missing. Especially the authors want to stress the stability of their materials, similar experiments in Figure 8 should be presented.
  14. Figure 7 need to be replotted. It looks like raw data. The reaction condition is not clearly presented for Figure 7, for example, the final concentration of AO7 and irradiation time.

Reviewer 3 Report

This is an scientifically interesting paper. However, we need to know the mechanism of the azo dyes degradation and the kinetic parameters of the degradation, namely rate constants. Moreover, this information must be compared with other works to justify the quality of the proposed methodology. 

Round 2

Reviewer 2 Report

The authors answered all the questions well.
The current version is good to go

Reviewer 3 Report

The authors have satisfactorily answered to the referee questions.